# Interactions of Different Species of *Phytophthora* with Cacao Induce Genetic, Biochemical, and Morphological Plant Alterations

**DOI:** 10.3390/microorganisms11051172

**Published:** 2023-04-29

**Authors:** Angra Paula Bomfim Rêgo, Irma Yuliana Mora-Ocampo, Ronan Xavier Corrêa

**Affiliations:** 1Centro de Biotecnologia e Genética (CBG), Universidade Estadual de Santa Cruz (UESC), Rodovia Jorge Amado km 16, Ilhéus 45662-900, Bahia, Brazil; angrabiotecnologista@gmail.com (A.P.B.R.); yulimoraocampo@gmail.com (I.Y.M.-O.); 2Departamento de Ciências Biológicas (DCB), Universidade Estadual de Santa Cruz, Ilhéus 45662-900, Bahia, Brazil

**Keywords:** plant disease, proteomics, plant defense, gene expression, resistance, systematic review, oomycetes, pathogen, pathogenesis, quantitative trait loci

## Abstract

Diseases associated with *Phytophthora* cause considerable losses in cocoa production worldwide. Analyzing genes, proteins, and metabolites involved in *Theobroma cacao*’s interaction with *Phytophthora* species is essential to explaining the molecular aspects of plant defense. Through a systematic literature review, this study aims to identify reports of genes, proteins, metabolites, morphological characteristics, and molecular and physiological processes of *T. cacao* involved in its interaction with species of *Phytophthora*. After the searches, 35 papers were selected for the data extraction stage, according to pre-established inclusion and exclusion criteria. In these studies, 657 genes and 32 metabolites, among other elements (molecules and molecular processes), were found to be involved in the interaction. The integration of this information resulted in the following conclusions: the expression patterns of pattern recognition receptors (PRRs) and a possible gene-to-gene interaction participate in cocoa resistance to *Phytophthora* spp.; the expression pattern of genes that encode pathogenesis-related (PRs) proteins is different between resistant and susceptible genotypes; phenolic compounds play an important role in preformed defenses; and proline accumulation may be involved in cell wall integrity. Only one proteomics study of *T. cacao-Phytophthora* spp. was found, and some genes proposed via QTL analysis were confirmed in transcriptomic studies.

## 1. Introduction

Diseases caused by species of oomycetes of the genus *Phytophthora* are responsible for substantial losses in the worldwide production of cocoa (*Theobroma cacao* L.) [1]. Among the most prevalent species, *P. palmivora* (E. J. Butler) affects cocoa plantations globally, while *P. megakarya* (Brasier and Griffin), which occurs only in Africa, affects the production of this cultivar. Other species such as *P. megakarya* [2], *P. capsici* (Leonian, 1922), and *P. citrophthora* (R.E. Sm. & E.H. Sm. Leonian) cause significant losses in cocoa production in Central and South America [1]. Furthermore, a new species, *P. theobromicola*, was recently discovered in cocoa beans collected from different farms in Bahia, Brazil [3].

The typical symptom caused by *Phytophthora* spp. is the rotting of the fruit’s skin. This rot starts with small, hard, dark lesions that overgrow in a few days, covering the entire fruit surface and internal tissues, including the kernels [4]. Other parts of the plant may also be affected, although less frequently, such as the stem, flower cushions, leaves, and roots [5].

The control of diseases caused by *Phytophthora* spp. mainly involves cultural techniques for disinfecting tools and applying chemicals. However, due to factors such as the high cost of labor and fungicides, and their impact on the environment, the use of cocoa varieties genetically resistant to *Phytophthora* is considered one of the most efficient forms of control [5,6].

The plant–pathogen interaction must be understood at several levels to develop tools that support disease control. At the phenotype level, the specificity and intensity of symptoms should be investigated, together with histological and morphological characteristics. Observing the symptoms makes it possible to identify which varieties are resistant and susceptible to the pathogen. In studying the interaction between *T. cacao* and *Phytophthora* spp., artificial inoculation techniques were developed and used to evaluate the susceptibility of the different cocoa organs in the laboratory and the field [5]. As a result, many studies evaluated the level of resistance of different cocoa varieties to *Phytophthora* spp. and subsequently identified different genotypes resistant to three species of *Phytophthora*: *P. palmivora*, *P. citrophthora*, and *P. capsici* [6].

At the gene level, it is important to know the indicators that differentiate genotypes with different degrees of resistance and how resistance is inherited to create resistant varieties [7]. In two genetic maps of *T. cacao*, quantitative trait loci (QTLs) were associated with brown rot resistance for the species *P. palmivora*, *P. megakarya*, *P. capsici*, and *P. citrophthora* [8,9].

Several studies have related genetic information with the phenotypic characteristics of diseases caused by *Phytophthora* spp. [9,10,11,12,13,14,15,16,17,18,19,20]. Based on these studies, it was possible to infer that *T. cacao* resistance to *Phytophthora* is quantitative or polygenic; that is, it exhibits continuous phenotypic variations, suggesting that many genes, each with small effects, contribute to the expression of cacao resistance to this pathogen [9,21].

Studies analyzing which genes, proteins, and metabolites are differentially expressed between healthy and infected plants and between varieties with different degrees of resistance are critical to discovering the molecular aspects of the interaction of *T. cacao* with *Phytophthora*.

Through a sensitive and broad search in the scientific literature, this study aims to identify and analyze the primary studies that identify or report genes, proteins, metabolites, morphological characteristics, and molecular and physiological processes of *T. cacao* involved in its interaction with the *Phytophthora* species.

## 2. Materials and Methods

### 2.1. String Construction and Selection for Database Searches

The objective of the systematic literature review was used as a basis to construct the main question and conduct the searches. The main question of this study is: “Which genes, proteins, metabolites, morphological characteristics, and molecular and physiological processes of *T. cacao* are involved in its interaction with *Phytophthora* sp.?”

From this central question, keywords were selected to construct the strings (Table 1), which were tested in the database for academic research Scopus (https://www.scopus.com, accessed on 1 January 2023). The string that would be used in other databases for academic research was selected according to the following criteria: (a) the number of articles considered useful for answering the main question according to the abstract of the first 20 articles that appeared in the list; (b) whether or not sentinel articles were included in the first 20 articles that appeared in the list; and (c) the specificity of the string (Table 1).

### 2.2. Literature Search

The string selected and used in the search was {TITLE-ABS-KEY[(cacao OR cocoa) AND (resistance OR responses OR interaction OR defense OR protein* OR gene*) AND (*Phytophthora*)]}. In some cases, this string was adapted according to the rules of each site for literature searches (full articles, reviews, short communications, scientific notes). The search was carried out in the academic databases Scopus (https://www.scopus.com, accessed on 1 January 2023), Web of Science (https://www.webofknowledge.com, accessed on 1 January 2023), Science Direct (https://www.sciencedirect.com, accessed on 1 January 2023), PubMed (https://pubmed.ncbi.nlm.nih.gov, accessed on 1 January 2023), and Scielo (https://scielo.org, accessed on 1 January 2023). The string keywords were translated into Spanish and Portuguese for the latter database. Only articles from journals indexed in scientific databases were selected.

### 2.3. Literature Selection

Initially, a protocol was filled out with the key questions that guided the research (Appendix A). The identification, selection, and data extraction stages were performed using the tool StArt (State of the Art through systematic review) [26] v.2.3.4.2. For the selection stage, inclusion and exclusion criteria were established, and the option to automatically identify duplicate articles was used. Inclusion criteria: studies addressing the molecular, biochemical, physiological, and/or morphological responses of cocoa to any species of *Phytophthora*; studies reporting genes, transcripts, proteins, and/or metabolites identified in cocoa in response to interaction with any species of *Phytophthora*; studies addressing the characterization of *T. cacao* genes or proteins, but in transgenic plants (whether or not *T. cacao*) that demonstrate response or resistance to *Phytophthora* sp. Exclusion criteria: studies reporting populations, accessions, and/or genotypes with resistance to brown rot, but ignore the analysis methodologies that explain resistance; studies addressing only *Phytophthora* sp., but relate resistance to another phytopathosystem; studies addressing only the genetics, proteomics, and/or physiology of *Phytophthora* sp. without relating them to cocoa; studies reporting only QTLs or molecular markers in cocoa related to resistance to *Phytophthora* sp., but without reporting specific genes; studies reporting methodologies for screening resistance of *T. cacao* to *Phytophthora* sp; studies addressing the interaction of *T. cacao* with any species of *Phytophthora*; studies addressing the influence of external treatments (biological or chemical) on the interaction between *T. cacao* and *Phytophthora* sp.

### 2.4. Data Extraction

Once the articles were selected according to the inclusion and exclusion criteria, the data extraction stage was performed. For this purpose, blanks to fill in were established, corresponding to the general information about each study, as shown in Table 2. For the blanks “type of analysis” and “plant tissue studied”, the options were fixed, and more than one option could be selected. In the blank “species of *Phytophthora* used in the study” and for each of the “reported elements”, a space for free text was provided. In addition, blanks of elements reported for both resistant and susceptible genotypes were made available. The year of publication and the country where the study was conducted were also extracted.

## 3. Results

### 3.1. Subsection Literature Search and Selection

A total of 490 articles were returned using the refined string in the five academic databases. About 80% of these were returned by Web of Science and Scopus, and the rest by the other three chosen databases (Figure 1A).

Of the total number of articles found, more than half were rejected to move on to the information extraction stage, according to the exclusion criteria (Figure 1B,C). Most were rejected because they do not address the interaction of *T. cacao* with some species of *Phytophthora* and only report populations/accessions/genotypes with resistance to brown rot, disregarding molecular analyses. Moreover, they only addressed the genetics, proteomics, or physiology of *Phytophthora* spp. without relating them to cocoa (Figure 1C).

In turn, more than 40% of the studies were detected as duplicates, and only 35 were accepted for the data extraction stage, according to the inclusion criteria (Figure 1B). Most accepted articles address the molecular, biochemical, physiological, and/or morphological responses of *T. cacao* to any *Phytophthora* spp.

### 3.2. Countries of Origin and Year of Publication of Papers Accepted for Data Extraction

Most papers accepted were developed in African countries and the USA (Figure 2). In South America, the country with the most articles published within the inclusion criteria was Trinidad and Tobago, followed by Brazil. More than 60% of the articles were published between 2011 and 2022 (Figure 2); the oldest was published in 1961.

### 3.3. Data Extraction

In the information extraction stage, blanks were established to be completed with the general attributes for each accepted article (Figure 3). Regarding the species of *Phytophthora*, the most widely selected to study the interaction with *T. cacao* were *P. megakarya* and *P. palmivora*. In turn, the most examined tissue was the leaf, followed by the fruit. In most accepted studies, the reported types of *T. cacao* elements involved in the interaction with *Phytophthora* sp. were genes and metabolites. Finally, regarding the type of analyses, the majority were on biochemistry, followed by genetics and transcriptomics. Only one study on proteomics [24] was within the inclusion criteria for articles accepted for data extraction.

### 3.4. Elements That Participate in the Interaction with Species of Phytophthora

Data were extracted from the twelve studies that reported genes involved in the interaction between cacao and *Phytophthora* sp. to create a table (Appendix A) with the 657 genes and 15 groups of transcripts of *T. cacao* considered the most relevant, most of which were related to defense.

Only one proteomics study on the interaction between cacao and *Phytophthora* sp. met the inclusion criteria [24]. This study identified 37 proteins in the resistant genotype and 39 proteins in the susceptible genotype through 2D-SDS-PAGE and mass spectrometry (Appendix A). Several of these proteins were identified more than once in different spots.

The other elements of *T. cacao* reported in the interaction with *Phytophthora* spp. are shown in Table 3. The most frequently reported enzyme was polyphenol oxidase. Almost all the secondary metabolites mentioned were phenolic compounds and their derivatives. In addition, four transiently expressed phosphatidylinositol-3-phosphate binding proteins were reported.

The identified molecular responses include signal transduction, response to various stresses and stimuli, and biosynthesis of phenylpropanoids, ethylene, and jasmonic acid. The reported morphological characteristics were almost always from the fruit of *T. cacao*, while only one study reported the morphological characteristics of the stem.

The only physiological response studied in the selected papers was moisture content. Other reported elements that possibly participate in the interaction between *T. cacao* and *Phytophthora* sp. were amino acids, lignin, carbohydrates, wax, caffeine, and synthetic peptides.

**Table 3 microorganisms-11-01172-t003:** Elements of *Theobroma cacao* that participate in the interaction with different species of *Phytophthora*.

Enzymes							
Enzyme activity	Level of activity	Time after inoculation	Effect on *T. cacao*	Methodology	Tissue studied	Species	References
Polyphenol oxidase	High	6 h	Resistance	Tissue maceration and dimming	Fruit/Beans	Ppal	[27]
Peroxidase	High	12 weeks	Resistance	Spectrophotometry	Stem	Ppal	[28]
Polyphenoloxidase	High	12 weeks	Resistance	Spectrophotometry	Stem	Ppal	[28]
Phenylalanine ammonium-lyase	High	12 weeks	Resistance	Spectrophotometry	Stem	Ppal	[28]
Polyphenoloxidase (isoforms)	High	6 days	Resistance	Spectrophotometry	Fruit	Pmeg	[29]
Metabolites							
Metabolite	Content	Time after Infection	Effect on *T. cacao*	Methodology	Tissue studied	Species	References
Tannins	High	6 h	Resistance	Tissue maceration and dimming	Fruit	Ppal	[27]
Flavonol	High	4 days	Resistance	Spectrophotometry	Fruit	Pmeg	[30]
Hydroxynamic derivatives	High	4 days	Resistance	Spectrophotometry	Fruit	Pmeg	[30]
Phenol	Presence	6 weeks	Resistance	Ferric chloride test	Stem	Ppal	[31]
Terpenoides	Absence	6 weeks	Resistance	Test of 2,4-DNP	Stem	Ppal	[31]
Glicosides	Absence	6 weeks	Resistance	Picric acid test	Stem	Ppal	[31]
Soluble phenolics compounds	High	6 days	Resistance	Spectrophotometry	Leaf	Pmeg	[32]
Luteolin derivatives	High	6 days	Resistance	HPLC	Leaf	Pmeg	[32]
Apigenin derivaties	High	6 days	Resistance	HPLC	Leaf	Pmeg	[32]
Derivatives of hydroxycincamic acids	High	6 days	Resistance	HPLC	Leaf	Pmeg	[32]
Soluble phenolic compounds	High	6 days	Resistance	Spectrophotometry	Leaf	Pmeg	[33]
Luteolin derivatives	High	6 days	Resistance	HPLC	Leaf	Pmeg	[33]
Apigenin derivatives	High	6 days	Resistance	HPLC	Leaf	Pmeg	[33]
Derivatives of hydroxycinnamic acids	High	6 days	Resistance	HPLC	Leaf	Pmeg	[33]
Soluble phenolic compounds	High	6 days	Resistance	Spectrophotometry	Leaf	Pmeg	[34]
Total polyphenols	High	6 days	Resistance	Folin-Ciocalteu reagent method	Leaf/Fruit	Pmeg	[7]
Flavonoides	High	6 days	Resistance	Aluminum chloride method	Leaf/Fruit	Pmeg	[7]
Tannins	High	6 days	Resistance	Vanillin method	Leaf/Fruit	Pmeg	[7]
Caffeoyl-DOPA (clovamide)	High	Basal	Resistance	LC-MS/MS	Leaf	Ppal	[22]
Coumaroyl-DOPA	High	Basal	Resistance	LC-MS/MS	Leaf	Ppal	[22]
Coumaroyl-Tyrosine	High	Basal	Resistance	LC-MS/MS	Leaf	Ppal	[22]
Sinapoyl-Tyrosine	High	Basal	Resistance	LC-MS/MS	Leaf	Ppal	[22]
Caffeoyl-Tryptophan	High	Basal	Resistance	LC-MS/MS	Leaf	Ppal	[22]
Caffeoyl-DOPA, Alkyl-Sulfated	High	Basal	Resistance	LC-MS/MS	Leaf	Ppal	[22]
Feruloyl-DOPA	High	Basal	Resistance	LC-MS/MS	Leaf	Ppal	[22]
Feruloyl-DOPA, Aryl-Sulfated	High	Basal	Resistance	LC-MS/MS	Leaf	Ppal	[22]
Caffeoyl-Phenylalanine	High	Basal	Resistance	LC-MS/MS	Leaf	Ppal	[22]
Caffeoyl-Tyrosine, Alkyl-Sulfated	High	Basal	Resistance	LC-MS/MS	Leaf	Ppal	[22]
Clovamide	Decrease	72 h	Resistance	LC-MS/MS	Fruit	Ppal	[22]
Aryl-sulfatedclovamide	Decrease	72 h	Resistance	LC-MS/MS	Fruit	Ppal	[22]
Feruloyl-DOPA	Decrease	72 h	Resistance	LC-MS/MS	Fruit	Ppal	[22]
Arylsulfatedferuloyl-DOPA	Decrease	72 h	Resistance	LC-MS/MS	Fruit	Ppal	[22]
Proteíns							
Protein name	Regulation	Time after inoculation	Resistance	Methodology	Tissue studied	Species	References
Vacuolar morphogenesis protein 7 (domain type: VAM7p-PX)	Expression	3 days	Resistance	Transiente transformation/Western blot	Leaf	Ptrop/Ppal	[35]
Hepatocyte growth factor-regulated tyrosine kinase substrate (domain type: Hrs-2xFYVE)	Expression	4 days	Resistance	Transiente transformation/Western blot	Leaf	Ptrop/Ppal	[35]
Pleckstrin homology domain-containing family A member 4 (domain type: PEPP1-PH)	Expression	5 days	Resistance	Transiente transformation/Western blot	Leaf	Ptrop/Ppal	[35]
PH domain-containing protein (domain type: GmPh1-PH)	Expression	6 days	Resistance	Transiente transformation/Western blot	Leaf	Ptrop/Ppal	[35]
List of differentially accumulated proteins (Appendix A)	Differential	48 h	Resistance	LC-MS/MS	Leaf	Ppal	[24]
Molecular Responses							
Responses	Regulation	Time after inoculation	Resistance	Methodology	Tissue studied	Species	References
Signal transduction inducing a gene defense response	Induced	NS	Resistance	Macroarray	Leaf	Pmeg	[36]
Biosynthesis of phenylpropanoids	Induced	24, 48 e 72 h	Resistance	RNA-Seq/RT-qPCR/KEGG	Leaf/Fruit	Ppal/Pmeg	[37]
Biosynthesis and action of ethylene	Induced	24, 48 e 72 h	Resistance	RNA-Seq/RT-qPCR/KEGG	Leaf/Fruit	Ppal/Pmeg	[37]
Biosynthesis and action of jasmonic acid	Induced	24, 48 e 72 h	Resistance	RNA-Seq/RT-qPCR/KEGG	Leaf/Fruit	Ppal/Pmeg	[37]
Defense signal transduction	Induced	24, 48 e 72 h	Resistance	RNA-Seq/RT-qPCR/KEGG	Leaf/Fruit	Ppal/Pmeg	[37]
Endocytosis	Induced	24, 48 e 72 h	Resistance	RNA-Seq/RT-qPCR/KEGG	Leaf/Fruit	Ppal/Pmeg	[37]
Response to stimuli and stress	Induced	24 h	Resistance	RNA-Seq/BLAST2GO	Leaf	Pmeg	[21]
Signaling	Induced	24 h	Resistance	RNA-Seq/BLAST2GO	Leaf	Pmeg	[21]
Morphological Characteristics							
Characteristics	Attribute	Time after inoculation	Resistance	Methodology	Tissue studied	Species	References
Shell thickness	Bigger	Basal	Resistance	Measuring with scale	Fruit	Ppal	[38]
Shell hardness	Bigge	2, 4, 6, 8, 10, 12 weeks	Resistance	Tensile/compression test	Root	Ppal	[39]
Epicarp thickness	Bigge	Basal	Resistance	Microscopy	Fruit	Ppal/Pmeg	[40]
Phloem fiber thickness	Bigge	Basal	Resistance	Microscopy	Fruit	Ppal/Pmeg	[40]
Length of vascular bundles	Bigge	Basal	Resistance	Microscopy	Fruit	Ppal/Pmeg	[40]
Distance between adjacent vascular bundles	Bigge	Basal	Resistance	Microscopy	Fruit	Ppal/Pmeg	[40]
Distance between vascular bundles and epicarp	Bigge	Basal	Resistance	Microscopy	Fruit	Ppal/Pmeg	[40]
Number of cells in the epicarp	Bigge	Basal	Resistance	Microscopy	Fruit	Ppal/Pmeg	[40]
Number of cells in the mesocarp	Bigge	Basal	Resistance	Microscopy	Fruit	Ppal/Pmeg	[40]
Number of vascular bundles	Smaller	Basal	Resistance	Microscopy	Fruit	Ppal/Pmeg	[40]
Stomatal frequency	Smalle	72 h	Resistance	Microscopy	Fruit	Ppal	[41]
Pore length	Smalle	72 h	Resistance	Microscopy	Fruit	Ppal	[41]
Physiological Responses							
Responses	Attribute	Time after inoculation	Resistance	Methodology	Tissue studied	Species	References
Moisture content	Bigger	2, 4, 6, 8, 10, 12 weeks	Susceptibility	Dry dough method	Root	Ppal	[39]
Moisture content	Variable	72 h	Unrelated	Dry dough method	Fruit	Ppal	[41]
Moisture content	Low	Basal	Resistance	Dry dough method	Fruit	Ppal	[38]
Others							
Compound	Content	Time after inoculation	Resistance	Methodology	Tissue studied	Species	References
Lignin	High	2, 4, 6, 8, 10, 12 weeks	Resistance	Phloroglucinol-HCl test	Root	Ppal	[39]
Proline	Increase	5 days	Resistance	Spectrophotometry	Fruit	Pmeg	[42]
Tyrosine	Increase	5 days	Resistance	Thin layer chromatography	Fruit	Pmeg	[42]
Aspartate	Increase	5 days	Resistance	Thin layer chromatography	Fruit	Pmeg	[42]
Soluble carbohydrates	Decrease	5 days	Resistance	Spectrophotometry	Fruit	Pmeg	[42]
Proline	Presence	6 days	Resistance	Thin layer chromatography	Leaf	Pmeg	[34]
Leucine	Presence	6 days	Resistance	Thin layer chromatography	Leaf	Pmeg	[34]
Sucrose	Absence	6 days	Resistance	Thin layer chromatography	Leaf	Pmeg	[34]
Synthetic peptides (US Patent # 5597945)	Expression	3 days	Resistance	Transgenic	Leaf	Pcap/Ppal	[43]
Wax	High	4 days	Resistance	Phenol wax wash	Leaf/Fruit	Ppal/Pmeg	[40]
Lignin	High	6 days	Resistance	Thioglycolic acid test	Leaf/Fruit	Pmeg	[7]
Totals proteins	Increase	6 days	Resistance	Nitrogen Content × 6.25	Leaf/Fruit	Pmeg	[7]
Soluble and insoluble sugars	Increase	6 days	Resistance	Phenol Sulfuric Method	Leaf/Fruit	Pmeg	[7]
Caffeine	High	NS	Resistance	RP-HPLC	Beans	NS	[44]

Ppal: *Phytophthora palmivora*; Pmeg: *Phytophthora megakarya*; *Phytophthora tropicalis*; Pcap: *Phytophthora capsici*.

## 4. Discussion

### 4.1. Few Primary Studies Address the Molecular and Morphological Aspects of the Interaction of T. cacao with Phytophthora spp.

Diseases associated with species of *Phytophthora* cause greater losses in cocoa production than any other disease [1]. Therefore, it is common for these species to be highlighted in abstracts of studies on cocoa in different areas, including its interaction with other pathogens. In turn, some studies address the morphological and molecular characteristics of species of *Phytophthora* and highlight the most important hosts, such as cacao. Thus, many articles were rejected by the exclusion criteria because they needed to address the interaction between *T. cacao* and *Phytophthora* (Figure 1C), although these keywords were included in the title or abstract.

The low number of articles accepted according to the inclusion criteria (35 out of 490 found via the keywords; Figure 1D) indicates that there are few primary studies analyzing the molecular and morphological aspect of the interaction of *T. cacao* with *Phytophthora* spp. Additionally, this reflects the availability of analysis technologies over time, as the number of articles that met the inclusion criteria doubled from 2002 to 2020 (Figure 2). During these years, technologies for molecular analysis, such as sequencing techniques, evolved rapidly and became more accessible over time. Moreover, the cocoa genome was sequenced and published only in 2010 [10], which allowed the advancement of research in the area of genetics and transcriptomics with this organism.

The two species most widely addressed in the study of interaction with cocoa in the articles accepted for data extraction were *P. palmivora* and *P. megakarya* (Figure 3). These species were probably chosen because *P. palmivora* causes huge losses worldwide, and *P. megakarya* affects the main cocoa-producing countries in Africa, the continent with the highest cocoa production in the world [1]. This also explains why African countries have developed more research on the biochemical, genetic, transcriptomic, and morphological interaction between cocoa and *Phytophthora* spp., together with the USA (Figure 2), which is a global leader in science and technology [45].

Regarding the type of analysis, most of the accepted articles were in biochemistry (Figure 3). This may be due to the accessibility and speed of the tests required for biochemical analysis. Moreover, it provides relevant information on metabolic pathway products’ role in plant defense. However, despite having more biochemical studies among the accepted works, there were only 32 records of metabolites (mainly phenolic compounds) and only 5 of enzyme activities. On the other hand, more than 600 records of genes were involved in the interaction of *T. cacao* with *Phytophthora* spp. (Table 3 and Appendix A), which highlights the importance of complex gene regulation processes in the plant’s defense response.

Although the fruit is part of the plant most affected by *Phytophthora* spp. [4], there were more records of the use of leaves in the studies accepted for data extraction (Figure 3). Leaf bioassays have allowed the early and rapid selection of cocoa cultivars resistant to brown rot [18]. Therefore, leaf assays are a faster and more efficient way to study the effects of the pathogen on the cacao tree.

### 4.2. Expression Patterns of Pattern Recognition Receptors (PRRs) Are Crucial in Cacao Resistance to Phytophthora spp.

In all, 81 different *T. cacao* genes encoding probable disease-resistance proteins have been reported (Appendix A) [21,36,37,46]. These genes were classified as possible pattern recognition receptors (PRRs) [21]. Among these, five genes had their expression repressed 24 h after inoculation with *P. megakarya*, in genotypes showing resistance, and one had its expression induced in the susceptible genotype 72 h after inoculation.

The gene expression before inoculation in resistant and susceptible genotypes of *T. cacao* was compared to identify genes involved in basal defense [21]. As a result, 20 genes encoding different probable defense-associated receptors were expressed in greater amounts in the resistant genotype than in the susceptible genotype. In the susceptible genotype, however, 27 genes that encode different probable receptors associated with defense were expressed in greater quantity when compared to the resistant genotype. Interestingly, only the susceptible genotype exhibited the leaf rust 10 disease-resistance loci receptor-like protein kinase-like sort, which is involved in abscisic acid (ABA) signaling [47], and only the resistant genotype exhibited inactive receptors (Appendix A).

PRRs are known to be the first step of the plant defense response, as they recognize molecular patterns of the pathogen or cell wall damage [48]. Different expression patterns of PRRs were observed between the resistant and susceptible genotypes [21]. These differences emphasize the importance of the differentiated set of receptors for pathogen recognition in the plant, as the pathogen can secrete effectors that block or prevent recognition by certain PRRs [48].

### 4.3. Expression Patterns of Genes Encoding Pathogenesis-Related (PRs) Proteins Differ between Resistant and Susceptible Genotypes

In all, 109 genes encoding pathogenesis-related (PRs) proteins were reported. These show different expression patterns according to characteristics such as (i) the degree of resistance of the *T. cacao* variety, (ii) the time after infection, and (iii) the studied species of *Phytophthora* (Appendix A) [8,21,23,36,37,49,50].

The genes in the cocoa genotype resistant to *P. palmivora* had their expression reduced 72 h after inoculation (HAI) and encoded the following proteins: two PR-2 (glucosidases), three PR-7 (subtilisin-like proteases), one PR-9 (peroxidase), and one PR-16 (pentatricopeptide repeat-containing protein) [49]. In turn, genes in the genotype resistant to *P. megakarya* had their expression reduced and encoded two PR-9 (24 HAI) and one PRB1-2 (72 HAI) [21].

The genes coding for PR proteins that had increased expression in the resistant genotype after inoculation with *P. megakarya* were two PR-2 (24 HAI), three PR-3 (24 HAI), two PR-6 (Glu *S. griseus* protease inhibitors; one at 24, 48, 72, and 96 HAI and the other with an unspecified time of inoculation), and two PR-9 (24 HAI) [21,36].

PR proteins are also called defense-related inducible proteins [51]. Signaling molecules induce PR in the cell, such as salicylic acid, jasmonic acid, and ethylene, after the pathogen has entered the cell. In a cacao–pathogen interaction research [49], microarrays were used to evaluate the gene expression level of the 17 families of PRs after inoculation with *P. palmivora* in a variety of *T. cacao* considered resistant to brown rot by [41]. In the study, 76 genes encoding PRs were induced. The family with the highest number of induced genes was PR-9, with twelve peroxidases, followed by PR-10 (eight ribonucleases), and PR-3 (eight chitinases).

However, the ability of PRs to prevent or reduce disease in plants depends on several factors, such as plant and pathogen species and the type of protein [51]. Inducing a large number of PRs does not imply that the plant will develop resistance to the pathogen. For example, the brown rot susceptible genotype had more PR family members induced after treatment with salicylic acid [52]. Interestingly, the biosynthesis and action routes of ethylene and jasmonic acid were induced in the *Phytophthora* spp. [37]. As discussed above, the compounds induce PR expression.

Expression of PRs also varies according to tissue and time of infection, where the expression of genes encoding a PR-1 changed depending on the tissue, with increased expression in the leaf and decreased expression in the fruit at 24 HAI [37].

### 4.4. Phenolic Compounds Play an Essential Role in Preformed Defenses

Baseline differences were observed in the gene expression of resistant and susceptible genotypes of cocoa inoculated with *P. megakarya* [21]. More dramatically expressed genes in the resistant genotype include Anthocyanidin 3-O-glucosyltransferase 5, which is involved in the biosynthesis of anthocyanin-containing compounds. Anthocyanins are phenolic compounds that may play an important role in plant defense against various pathogens [53]. Another gene with a higher basal expression in the resistant genotype encodes a detoxification protein [42]. This protein is also involved in the biosynthesis of phenolic compounds (flavonoids and proanthocyanidins) [54].

In this regard, phenolic compounds were the most reported metabolites in accepted papers that studied the *T. cacao* and *Phytophthora* spp. (Table 3). The profiles of basal metabolites in cocoa genotypes resistant and susceptible to *Phytophthora* species show essential differences. The resistant genotype accumulated higher levels of clovamide and several other amides of hydroxycinnamic acid (HCAAs) compared to the susceptible genotype, with clovamide being the most abundant metabolite [22].

This information suggests that the basal content of phenolic compounds and the capacity to synthesize these compounds in the initial stage of the infection are critical to the resistance of *T. cacao* to *Phytophthora* spp.

### 4.5. Possible Gene-to-Gene Interaction May Be a Factor in the Resistance of T. cacao to Phytophthora spp.

According to previous studies, no gene-to-gene interactions have been demonstrated between cocoa and any of its pathogens [21]. However, a gene encoding a probable disease-resistance protein containing the NB-ARC domain was identified with much higher basal expression in the resistant genotype compared to the susceptible genotype ([21]; Appendix A). In potatoes, this type of protein provides resistance to *Phytophthora infestans* carrying the Avr1 avirulence gene [55,56]. Therefore, this gene containing NB-ARC may be fundamental in the incompatible interaction of the resistant cacao genotype with some pathotypes of *Phytophthora* spp.

### 4.6. Genes Proposed via QTL Analysis Were Confirmed in Transcriptomic Studies

Some genes close to the QTL regions in cacao associated with resistance to *P. palmivora*, *P. citrophthora*, and *P. capsici* were discovered using chromosomal locations of the markers as a reference [8]. Interestingly, several genes proposed in this study were identified through transcriptomic analyses in other studies (Appendix A), as were genes that encode glutaredoxin-C11 [21], calcium-binding protein CML23 [21], an F-box protein [21], an F-box/LRR-repeat protein [21], an LRR receptor-like serine/threonine-protein [21,37], a peroxidase [21,49], a serine-threonine protein kinase [21], and a zinc finger CCCH domain-containing protein [21]. However, the localized genes shown previously in the QTL regions do not contain the genome locus ID. When looking for the indicated location on each chromosome, the genes found in the cacao genome bank do not coincide with the gene name in the table, possibly due to a recent update to these loci.

A similar situation was identified in the other study [46], where three of the 324 candidate genes for recognition and activation of responses to the pathogen, identified in 20 QTL regions associated with resistance to *P. palmivora*, coincide with genes identified as differentially expressed [21], namely Tc10v2_g012100 (putative cc-nbs-lrr resistance protein), Tc10v2_g012750, and Tc10v2_g012770 (putative receptor-like protein 12). The two genes corresponding to receptor-like protein 12 were reported to be up-regulated in the resistant genotype compared to the susceptible genotype before inoculation with *P. megakarya*. In turn, the gene corresponding to the putative CC-NBS-LRR resistance protein was down-regulated in the resistant genotype 24 h after inoculation with *P. megakarya* [21].

### 4.7. Accumulation of Proline May Be Involved in Cell Wall Integrity

Proline is an amino acid that accumulates under stress and normal conditions as a beneficial solute in plants and plays an essential role in cell wall proteins [57]. Proline stood out as a differential element between resistant and susceptible genotypes in two studies ([34,42]; Table 3).

Furthermore, a gene encoding an ornithine decarboxylase, involved in the biosynthesis of a proline precursor [58], was reported to be induced at 48 HAI with *P. megakarya* ([59]; Appendix A). Regarding its importance in cell wall proteins such as arabinogalactan-proteins (AGPs), a gene encoding a Fasciclin-like arabinogalactan protein 2, rich in proline, was reported to be induced, 72 HAI, in a genotype resistant to *P. megakarya* ([21]; Appendix A). This information suggests that the induction of proline accumulation in resistant genotypes may be involved in the rapid recovery of the cell wall, which would be a factor in cocoa resistance to *Phytophthora* spp. and is consistent with the high lignin content found in resistant genotypes ([7,39]; Appendix A).

### 4.8. A Study at the Level of Proteomics Reveals Crucial Proteins in the Interaction of Cacao and P. palmivora

Despite the great economic losses due to diseases caused by *Phytophthora* spp. in *T. cacao*, only one study involving the proteomic analysis of this phytopathosystem was found (Figure 3; [24]). In the study, the proteomic profile of leaves of the genotypes of *T. cacao* PA150 (resistant) and SIC23 (susceptible) 48 HAI with the pathogen *P. palmivora* was compared with the respective controls inoculated with sterile distilled water [24]. The lipoxygenase proteins, 2-methylene-furan-3-one, co-chaperonin CPN20, and a probable CC-NBS-LRR stand out for having differentiated accumulation in the inoculated resistance genotype and are probably involved in the resistance of cocoa to *P. palmivora*. Interestingly, a gene encoding a CC-NBS-LRR was identified as expressed basally in the susceptible genotype [21]. In our analysis, this protein had a different accumulation pattern between the resistant and susceptible genotypes.

Among the transcriptomic studies reported in the present study, only one confirmed the expression of proteins through the Western blot technique [35]. Among these proteins, the four phosphatidylinositol-3-phosphate binding proteins, transiently expressed in *T. cacao* leaves, reduced infection by *P. tropicalis* and *P. palmivora* due to their ability to block the effector’s entry of the pathogen to the cell. Despite the abundance of data about genes differentially expressed in cocoa before and during its interaction with *Phytophthora* spp., studies at the proteomic level are especially relevant since gene expression is regulated at different stages through processes that occur after the synthesis of transcripts such as post-transcriptional, translational, post-translational regulation, and protein degradation [60,61].

## 5. Conclusions

Expression patterns of pattern recognition receptors (PRRs) and a possible gene-for-gene interaction participate in cacao resistance to *Phytophthora* spp.

The expression pattern of genes encoding pathogenesis-related proteins (PRs) is different between resistant and susceptible genotypes.

Phenolic compounds play an important role in preformed defenses.

Proline accumulation may be involved in cell wall integrity.

Genes proposed via QTL analysis encoding a glutaredoxin-C11, a calcium-binding protein CML23, an F-box protein, an F-box /LRR-repeat protein, an LRR receptor-like serine/threonine protein, a peroxidase, a serine-threonine protein kinase, and a zinc finger CCCH domain-containing protein were confirmed in transcriptomic studies.

## Figures and Tables

**Figure 1 microorganisms-11-01172-f001:**
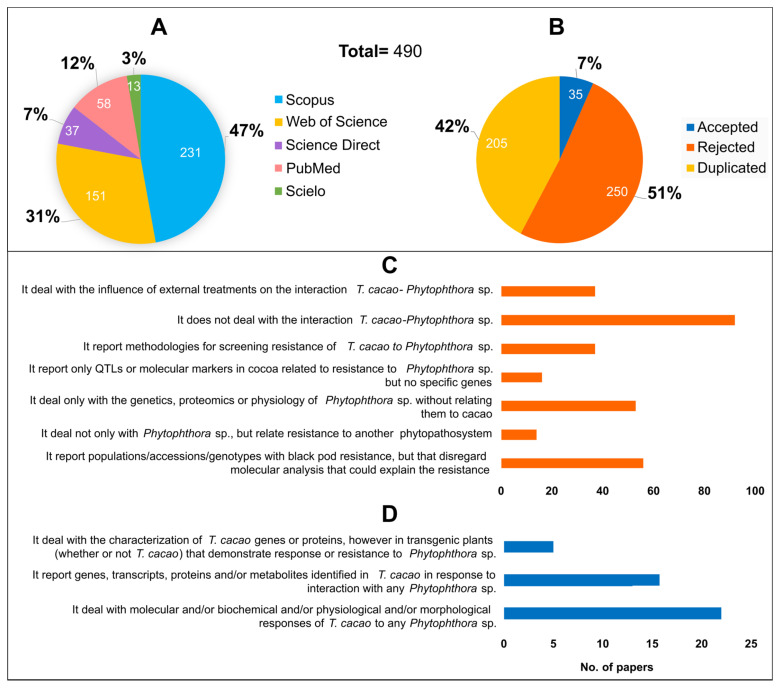
Articles found, accepted, and rejected for the extraction stage. (**A**) Articles found in each academic database using the refined string; (**B**) ranking of the article selection stage before data extraction; (**C**) number of articles rejected according to the exclusion criteria; (**D**) number of articles accepted according to the inclusion criteria.

**Figure 2 microorganisms-11-01172-f002:**
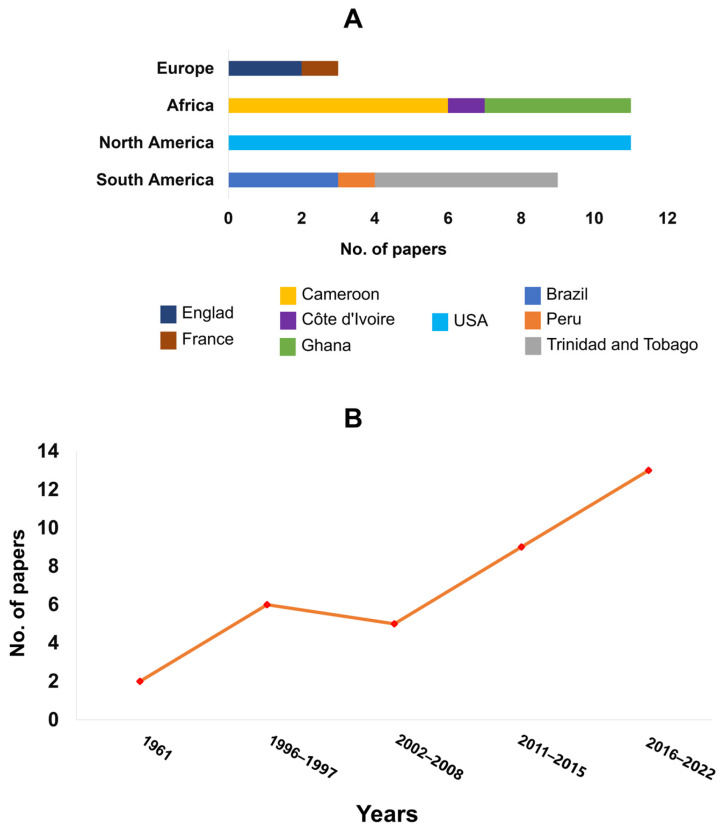
Countries of origin (**A**) and year (**B**) of published articles accepted according to the inclusion criteria.

**Figure 3 microorganisms-11-01172-f003:**
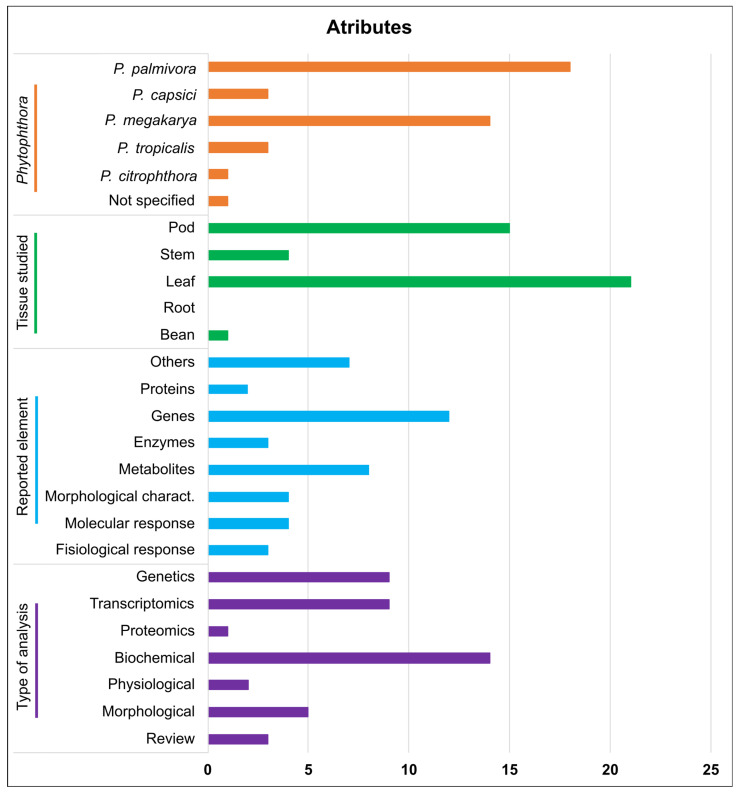
Enzymes, metabolites, proteins, and other elements that participate in the interaction of *Theobroma cacao* with species of *Phytophthora*.

**Table 1 microorganisms-11-01172-t001:** String refinement for searches in academic databases according to the main question of the study and sentinel articles, selecting the title, abstract, and keyword fields (TITLE-ABS-KEY).

String Tested on Scopus ^a^	Found Articles	Useful Articles for Answering the Question	Sentinel Articles in the First 20 Articles on the List ^b^
(cacao OR cocoa) AND (*Phytophthora*)	819	2	2
((cacao OR cocoa) AND (*Phytophthora*))	385	3	2
((cacao OR cocoa) AND (*Phytophthora*) AND (resistance OR response))	145	4	3
((cacao OR cocoa) AND (resistance OR response* OR interaction OR defense) AND (*Phytophthora*))	165	4	3
**((cacao OR cocoa) AND (resistance OR response* OR interaction OR defense OR protein* OR gene*) AND (*Phytophthora*))**	231	4	4

^a^ Tested strings in Scopus (https://www.scopus.com, accessed on 1 January 2023) for refinement by selecting the title, extract, and keyword fields (TITLE-ABS-KEY). ^b^ Sentinel articles Clovamide, a Hydroxycinnamic Acid Amide, Is a Resistance Factor Against *Phytophthora* spp. in *Theobroma cacao* [22], Developmental expression of stress response genes in *Theobroma cacao* leaves and their response to Nep1 treatment and a compatible infection by *Phytophthora megakarya* [23], Protein Level Defense Responses of *Theobroma cacao* Interaction with *Phytophthora palmivora* [24] and Changes in Gene Expression in Leaves of Cacao Genotypes Resistant and Susceptible to *Phytophthora palmivora* Infection [25] present in the first 20 items in the list). The string highlighted in bold was selected for the literature search. * = Boolean operator: the search engine will return any word that begins with the root/stem of the word truncated by the asterisk.

**Table 2 microorganisms-11-01172-t002:** Data extracted from the primary studies in interactions of *Theobroma cacao* and *Phytophthora* species.

Type of analysis	Genetic
	Transcriptomic
	Proteomic
	Biochemical
	Physiological
	Morphological
Plant tissue studied	Fruit
	Stem
	Leaf
	Root
	Seed
Reported element of *T. cacao* (resistant and susceptible) that participates in the interaction with species of *Phytophthora*	Protein
	Gene
	Enzyme
	Metabolite
	Morphological structure
	Molecular response
	Physiological response
	Others
Species of *Phytophthora*	
Country where he was accomplished the study	
Year of publication	

## Data Availability

The data presented in this study are inserted in the article or Appendix A.

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
