# Peer review of "Interactions of Different Species of *Phytophthora* with Cacao Induce Genetic, Biochemical, and Morphological Plant Alterations"

_microorganisms, 2023, doi:10.3390/microorganisms11051172_

Round 1

Reviewer 1 Report

This review summarized genes involved in the interactions between Phytophthora with cacao. They concluded that Phytophthora infections induce genetic, biochemical, and morphological changes in cacao at both transcriptional and translational levels. The literature search was well-performed and the manuscript was well-written. It may be published pending some moderate revisions.

1) Most functional genes encode proteins or enzymes. Therefore, Line 18 "three enzymes", and "proteins" and "enzymes" in Figure 3 and Table 4 are confusing/inaccurate categories.

2) In Line 18 "among other elements", and in Line 169 "3.4. Elements that participate in the interaction with species of Phytophthora", what elements?

3) In Figure 3, "review" is not a "type of study".

4) In Table 4, what are "Ppal" and "Pmeg" respectively?

5) Besides pathogenesis related (PRs) proteins, are there any other genes/proteins different between resistant and susceptible genotypes? which should be listed in a separate table.

6) In references, journal's names are shown in either full names or abbreviations.

Fine.

Author Response

We are grateful for the corrections and suggestions from reviewers. Corrections have been made to the article, as explained below.

We highlight the text in purple to indicate corrections requested by Reviewer 1, and in yellow to suggest corrections requested by Reviewer 2. Additionally, we highlight in green some suggestions we received from the first author's doctoral thesis defense committee.

We have combined the responses to all reviewers in this file because sometimes a modification refers to the same part of the text. Doe to the same reason, we show all corrections in a unified manuscript text.

Reviewer 1

"Comments and Suggestions for Authors. This review summarized genes involved in the interactions between Phytophthora with cacao. They concluded that Phytophthora infections induce genetic, biochemical, and morphological changes in cacao at both transcriptional and translational levels. The literature search was well-performed and the manuscript was well-written. It may be published pending some moderate revisions.

1) Most functional genes encode proteins or enzymes. Therefore, Line 18 "three enzymes", and "proteins" and "enzymes" in Figure 3 and Table 4 are confusing/inaccurate categories.

Authors' response: Line 18 text has been rewritten for clarity, making it more general in the abstract. In Table 4 (re-numbered to Table 3 due to the removal of Table 2 at the suggestion of another reviewer), now Table 3, we have made corrections to some terms.

2) In Line 18 "among other elements", and in Line 169 "3.4. Elements that participate in the interaction with species of Phytophthora", what elements?

Authors' response: we rewrote several parts of the work, giving examples of what we call elements to clarify this question. We call elements a set of proteins, metabolites, morphological characteristics and molecular processes identified in primary studies.

3) In Figure 3, "review" is not a "type of study".

Authors' response: We modified the following expressions: where “Type of study” was written, it was corrected to “Type of analysis.”

4) In Table 4, what are "Ppal" and "Pmeg" respectively?

Authors' response: in table 4, now renumbered to table 3, we have included the names of these two species in the table's title, indicating the meaning of the respective abbreviations.

5) Besides pathogenesis related (PRs) proteins, are there any other genes/proteins different between resistant and susceptible genotypes? which should be listed in a separate table.

Authors' response: We understand that the reviewer wanted to find a table with contrasting genes and proteins between resistant and susceptible in this article. However, this cannot be done based on the analysis of the articles, as the difference will depend on other factors such as the species of Phytophthora, time after infection, etc. The tables presented in this article are constructed so the reader can conclude according to the conditions he seeks. Thus, for the specific cases that a given researcher is studying, our study guides him to the subset of primary studies so that he can deepen comparisons pertinent to the particular cases.

6) In references, journal's names are shown in either full names or abbreviations."

Authors' Response: We adjusted all journal names to the full name.

Reviewer 2

Suggestions:

L12 and L15 Theobroma cacao should be italic.

Authors' Response: We've double-checked all occurrences of “* cacao” and italicized them.

L16-18: The results are very short in the Abstract compared tot he conclusions (L18-L25).

Authors' Response: The result of this work and other information related to this topic are detailed in the result itself. The emphasis on conclusions present in the abstract can attract more readers than the detailed results.

L99: Table 1 titole is not informative enough. Give more information in the title.

Authors' response: The Table 1 title has been rewritten to show the search string refinement strategy clearly.

Table 1. String refinement for searches in academic databases according to the central question of the study and sentinel articles, selecting the title, abstract, and keyword fields (TITLE-ABS-KEY).

L126: Table 2 is not a classical table. It is a text listing. This table should be deleted and information should included in the text.

Authors' response: We turned the information into a running paragraph.

L191: Give full latin name in the title of this Table.

Authors' Response: We include the full species names in the titles of tables and figures.

References: Journal name citation is inconsistent. Sometimes full name, sometimes short names provided.

Authors' response: we have corrected the journal names so that they are all in full name, as requested by reviewer 1.

Thesis Committee board:

The authors made additional modifications based on the suggestions of the first author's doctoral thesis evaluation board.

In the abstract, we corrected the number of 35 primary studies selected in this study, as it was incorrectly numbered.

At the end of the introduction, we added a few words to make explicit the use of “primary studies” and inform that they are multiple species of the genus Phytophthora.

At the beginning of item 2.3, we added an introductory sentence in the paragraph to announce that the protocol was presented as supplementary material, as is customary in systematic literature review studies.

We have made some corrections to Table 2 for clarity and to be in line with the nomenclature in Figure 3.

Figure 1 was modified about the number of accepted and duplicate articles before and after eliminating duplicate articles.

Figure 3 was modified, changing the category “Type of studies” to “Type of analysis”; the genus name has been italicized.

In Table 3, some terms, such as abbreviated words, have been corrected.

In item 4.3, there was a change in the order of a single sentence to restructure the paragraph better.

Professor Ronan Xavier Corrêa

Universidade Estadual de Santa Cruz

ronanxc@uesc.br

Author Response

Dear Reviewer
We appreciate your new reviews. All of them have been included. We modified the text in all the places that would involve mixing the two systems (numbered citation versus surname citation). Additionally, we changed the beginning of the objective to value the type of study.
You can find in the attachment the changes marked in word changes mode.
Thank you very much

Reviewer 3 Report

This study aims to identify in the literature the existence of primary studies that identify and/or report genes, proteins, metabolites, morphological characteristics, and molecular and physiological processes of cacao involved in its interaction with the species Phytophthora. The study design is acceptable. The study contains some interesting elements that can be published after suitable revisions.

Suggestions:

L12 and L15 Theobroma cacao should be italic.

L16-18: The results are very short in the Abstract compared tot he conclusions (L18-L25).

L99: Table 1 titole is not informative enough. Give more information in the title.

L126: Table 2 is not a classical table. It is a text listing. This table should be deleted and information should included in the text.

L191: Give full latin name in the title of this Table.

References: Journal name citation is inconsistent. Sometimes full name, sometimes short names provided.

Author Response

We are grateful for the corrections and suggestions from reviewers. Corrections have been made to the article, as explained below.

We highlight the text in purple to indicate corrections requested by Reviewer 1, and in yellow to suggest corrections requested by Reviewer 2. Additionally, we highlight in green some suggestions we received from the first author's doctoral thesis defense committee.

We have combined the responses to all reviewers in this file because sometimes a modification refers to the same part of the text. Doe to the same reason, we show all corrections in a unified manuscript text.

Reviewer 2

Suggestions:

L12 and L15 Theobroma cacao should be italic.

Authors' Response: We've double-checked all occurrences of “* cacao” and italicized them.

L16-18: The results are very short in the Abstract compared tot he conclusions (L18-L25).

Authors' Response: The result of this work and other information related to this topic are detailed in the result itself. The emphasis on conclusions present in the abstract can attract more readers than the detailed results.

L99: Table 1 titole is not informative enough. Give more information in the title.

Authors' response: The Table 1 title has been rewritten to show the search string refinement strategy clearly.

Table 1. String refinement for searches in academic databases according to the central question of the study and sentinel articles, selecting the title, abstract, and keyword fields (TITLE-ABS-KEY).

L126: Table 2 is not a classical table. It is a text listing. This table should be deleted and information should included in the text.

Authors' response: We turned the information into a running paragraph.

L191: Give full latin name in the title of this Table.

Authors' Response: We include the full species names in the titles of tables and figures.

References: Journal name citation is inconsistent. Sometimes full name, sometimes short names provided.

Authors' response: we have corrected the journal names so that they are all in full name, as requested by reviewer 1.

Reviewer 1

"Comments and Suggestions for Authors. This review summarized genes involved in the interactions between Phytophthora with cacao. They concluded that Phytophthora infections induce genetic, biochemical, and morphological changes in cacao at both transcriptional and translational levels. The literature search was well-performed and the manuscript was well-written. It may be published pending some moderate revisions.

1) Most functional genes encode proteins or enzymes. Therefore, Line 18 "three enzymes", and "proteins" and "enzymes" in Figure 3 and Table 4 are confusing/inaccurate categories.

Authors' response: Line 18 text has been rewritten for clarity, making it more general in the abstract. In Table 4 (re-numbered to Table 3 due to the removal of Table 2 at the suggestion of another reviewer), now Table 3, we have made corrections to some terms.

2) In Line 18 "among other elements", and in Line 169 "3.4. Elements that participate in the interaction with species of Phytophthora", what elements?

Authors' response: we rewrote several parts of the work, giving examples of what we call elements to clarify this question. We call elements a set of proteins, metabolites, morphological characteristics and molecular processes identified in primary studies.

3) In Figure 3, "review" is not a "type of study".

Authors' response: We modified the following expressions: where “Type of study” was written, it was corrected to “Type of analysis.”

4) In Table 4, what are "Ppal" and "Pmeg" respectively?

Authors' response: in table 4, now renumbered to table 3, we have included the names of these two species in the table's title, indicating the meaning of the respective abbreviations.

5) Besides pathogenesis related (PRs) proteins, are there any other genes/proteins different between resistant and susceptible genotypes? which should be listed in a separate table.

Authors' response: We understand that the reviewer wanted to find a table with contrasting genes and proteins between resistant and susceptible in this article. However, this cannot be done based on the analysis of the articles, as the difference will depend on other factors such as the species of Phytophthora, time after infection, etc. The tables presented in this article are constructed so the reader can conclude according to the conditions he seeks. Thus, for the specific cases that a given researcher is studying, our study guides him to the subset of primary studies so that he can deepen comparisons pertinent to the particular cases.

6) In references, journal's names are shown in either full names or abbreviations."

Authors' Response: We adjusted all journal names to the full name.

Thesis Committee board:

The authors made additional modifications based on the suggestions of the first author's doctoral thesis evaluation board.

In the abstract, we corrected the number of 35 primary studies selected in this study, as it was incorrectly numbered.

At the end of the introduction, we added a few words to make explicit the use of “primary studies” and inform that they are multiple species of the genus Phytophthora.

At the beginning of item 2.3, we added an introductory sentence in the paragraph to announce that the protocol was presented as supplementary material, as is customary in systematic literature review studies.

We have made some corrections to Table 2 for clarity and to be in line with the nomenclature in Figure 3.

Figure 1 was modified about the number of accepted and duplicate articles before and after eliminating duplicate articles.

Figure 3 was modified, changing the category “Type of studies” to “Type of analysis”; the genus name has been italicized.

In Table 3, some terms, such as abbreviated words, have been corrected.

In item 4.3, there was a change in the order of a single sentence to restructure the paragraph better.

Professor Ronan Xavier Corrêa

Universidade Estadual de Santa Cruz

ronanxc@uesc.br